DOI: 10.1038/s41467-017-01299-5　　**OPEN**

# Direct benefits explain interspecific variation in helping behaviour among cooperatively breeding birds

Sjouke A. Kingma [1]

Kin selection theory provides one important explanation for seemingly altruistic helping behaviour by non-breeding subordinates in cooperative breeding animals. However, it cannot explain why helpers in many species provide energetically costly care to unrelated offspring. Here, I use comparative analyses to show that direct fitness benefits of helping others, associated with future opportunities to breed in the resident territory, are responsible for the widespread variation in helping effort (offspring food provisioning) and kin discrimination across cooperatively breeding birds. In species where prospects of territory inheritance are larger, subordinates provide more help, and, unlike subordinates that cannot inherit a territory, do not preferentially direct care towards related offspring. Thus, while kin selection can underlie helping behaviour in some species, direct benefits are much more important than currently recognised and explain why unrelated individuals provide substantial help in many bird species.

---

[1] Behavioural & Physiological Ecology, Groningen Institute for Evolutionary Life Sciences, University of Groningen, P. O. Box 11103, Groningen 9700 CC, The Netherlands. Correspondence and requests for materials should be addressed to S.A.K. (email: s.a.kingma@rug.nl)

Approximately 9% of all bird species breed cooperatively, where non-reproducing subordinate "helpers" assist in raising the offspring of others[1,2]. Since helpers forego their own reproduction to provide energetically costly help[3], cooperative breeding has become a model system to study the major puzzle of how seemingly altruistic behaviour can remain evolutionarily stable[4,5]. One widely accepted adaptive explanation for helping behaviour is provided by kin-selection theory, which posits that if helpers assist relatives, they increase the trans-generational transfer of genes they share with the beneficiaries[6–10]. Comparative studies have highlighted the importance of kin selection for explaining variation in helping behaviour within[7,8] and across[9,10] cooperatively breeding bird species. However, on average only 10% of within-species variation in helping effort can be explained by variation in relatedness and, in many species, subordinates help non-relatives[7]. Clearly, kin selection alone cannot explain helping behaviour[11]. To understand the evolutionary maintenance of cooperative breeding, and cooperation and sociality more generally, we must determine within and across species: (i) the factors responsible for the widespread variation in helping behaviour, and (ii) the extent to which help is preferentially directed to more related individuals[5,12].

Direct fitness benefits associated with future reproduction are hypothesised to provide an additional mechanism underlying the evolution of cooperative breeding[4,5,13–15]. In many cooperative breeders, shortage of suitable territories (habitat saturation) limits subordinates' opportunities for independent reproduction[1,16] and theory predicts that both the lack of outside options and the prospects of territory inheritance may explain why such subordinates stay in a group and help[15,17–19]. Importantly, helping behaviour may facilitate survival and ultimate territory inheritance because helpers avoid aggression and eviction by breeders ("pay-to-stay" hypothesis[19,20]) or contribute towards the establishment of larger cooperative groups that improve survival, territory defence, group stability, or the ability to expand and split the territory ("group augmentation" hypothesis[5,15,21]). Despite this clear theoretical expectation and the fact that territory inheritance is a common and important route to independent breeding in many species[21], it remains unclear whether habitat saturation and prospects of territory inheritance can explain helping behaviour, especially by unrelated individuals who do not gain kin-selected benefits.

While territory shortage explains delayed independent breeding in many species, in others this is not the case, either because these species are not territorial, or because subordinates are sexually immature or are breeders who redirect their care towards the offspring of others when their own reproductive attempt fails[1,16]. This dichotomy between species with and without territory shortage provides the opportunity to test whether territory shortage (i.e., the lack of outside options), and thus the relative importance of inheriting the resident territory for future reproduction, explains helping by unrelated individuals and the widespread variation in helping behaviour, which is the aim of this comparative study. Specifically, if all territories in the population are occupied and helping promotes territory inheritance, subordinates should help regardless of whether or not they are related to the recipient of their help. To test this hypothesis, I collected data on helping effort (measured as average offspring provisioning rates of subordinates relative to breeders of the same sex[10]) and degree of kin discrimination (the species-specific correlation coefficient between relatedness and helping[7,8]) from published papers on 44 cooperatively breeding species. I subsequently compared these measures between species with territory shortage (i.e., species in which

independent breeding by subordinates is constrained by a shortage of vacant territories for independent breeding) and those without (i.e., colonial species, species with redirected care and species with immature helpers). The analyses reveal that prospects of territory inheritance are responsible for a large part of the currently unexplained variation in helping behaviour and kin discrimination (i.e., the extent to which helping behaviour is preferentially directed to more related individuals[7,8]) in cooperatively breeding birds.

## Results

**Territory shortage and helping behaviour.** In line with the expectations, I found that in species with territory shortage, on average ($\pm$SE) $30 \pm 4\%$ of subordinates inherit (part of) their territory (Supplementary Data 1). Subordinates in these species are considerably less discriminative in adjusting their investment based on kinship than subordinates in species without territory shortage (phylogenetic generalised least squares (PGLS) model: $t = -2.841$, $n = 21$ species, $P = 0.011$; Fig. 1a; Supplementary Table 1). Because subordinates contribute on average more to offspring provisioning in species that discriminate less (suggesting that unrelated subordinates help more in such species; PGLS: estimate $= -46.528 \pm 12.305$, $t = -3.781$, $n = 21$ species, $P = 0.001$), in species with territory shortage subordinates invest on average 51% more in offspring provisioning than subordinates in species where territory shortage does not constrain independent breeding (PGLS: $t = 2.764$, $n = 44$ species, $P = 0.009$; Fig. 1b; Supplementary Table 2a). In 12 species, some subordinates did not contribute to feeding offspring (Supplementary Data 1) and these were included in the calculation of helping effort. The results are, however, similar when these "non-helping" subordinates are excluded from the calculation of helping effort (Supplementary Table 2b). Moreover, since values of helping effort may be inflated if breeder males (included in the reference group to define helper effort) reduce their effort as a result of extra-pair mating by their social partner, I repeated the analyses correcting for rates of extra-pair paternity using a subset of species for which this was known. However, including extra-pair paternity rates did not change the results (Supplementary Table 2c).

For the analyses, species were categorised as species with and species without territory shortage. However, social systems vary considerably within these categories: species without territory shortage can be colonial, or involve juvenile helpers or redirected care, while those with territory shortage may involve either retained offspring or plural breeding systems where multiple females build nests in the same territory (see Supplementary Data 1 for details). The limited number of species with each of these social systems is not adequate to test for statistical differences in kin discrimination and helping effort, but, compared to species with territory shortage, the degree of kin discrimination was relatively high and helping effort was relatively low for species in which there is no territory shortage, regardless of the social system (kin discrimination (mean correlation coefficient $\pm$ SE): species with retained offspring: $0.14 \pm 0.10$, $n = 11$ species; plural breeding: $0.07 \pm 0.06$, $n = 2$ species; colonial: $0.49 \pm 0.10$, $n = 4$ species; redirected care: $0.57 \pm 0.20$, $n = 3$ species; see Supplementary Data 1). Helping effort (mean $\pm$ SE percentage offspring food provisioning per helper, relative to breeders): species with retained offspring: $89 \pm 10\%$, $n = 24$ species; plural breeding: $88 \pm 13\%$, $n = 4$ species; colonial: $63 \pm 7\%$, $n = 5$ species; redirected care: $56 \pm 13\%$, $n = 6$ species; immature helpers: $53 \pm 1\%$, $n = 2$ species (see Supplementary Data 1). This indicates that the reported overall effects were not driven by species with a

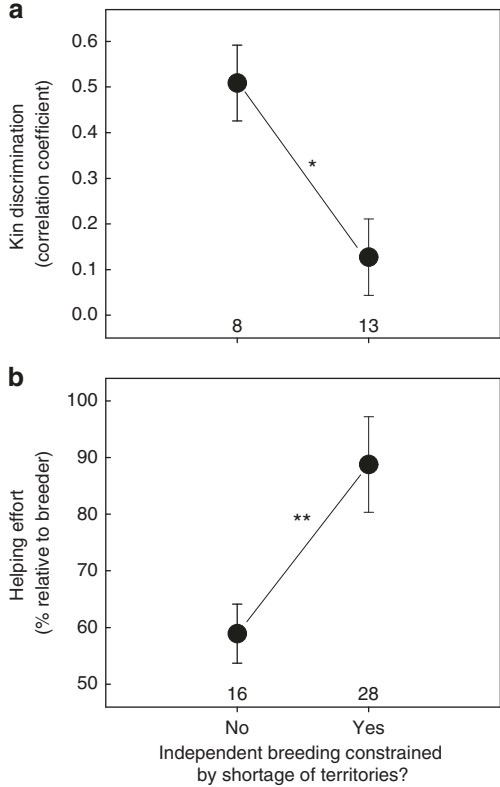

**Fig. 1** Territory shortage affects helping behaviour in cooperatively breeding birds. **a** Helpers in 8 species with no territory shortage mainly direct care to kin, whereas levels of kin discrimination are low (i.e., helpers do not provision kin more than non-kin) in 13 species in which a shortage of vacant territories constrains independent breeding (PGLS model: $P = 0.011$; the model output is provided in Supplementary Table 1). **b** As a result, helping effort (mean % offspring food provisioning per helper, relative to breeders) in the 28 species with territory shortage is higher than in the 16 species in which independent breeding is not constrained by territory shortage (PGLS model: $P = 0.009$; the model output is provided in Supplementary Table 2). Data points and errors bars show means ± standard errors. Numbers reflect the number of species. Asterisk and double asterisks reflect significant effects with $P < 0.05$ and $P < 0.01$, respectively

particular social system either for kin discrimination or for helping effort.

The mean coefficient of determination ($r^2$) between helping and relatedness in species with territory shortage was $0.10 \pm 0.03$ (range = 0.0004–0.36), meaning that within species, on average only 10% of the variation in helping is explained by relatedness (including four species in which unrelated subordinates actually provide more help than related subordinates). In contrast, in species that do not obtain direct benefits of territory inheritance, subordinates strongly adjust their investment towards related offspring: relatedness explains on average $31 \pm 10\%$ (range = 6–93%) of the variation in helping behaviour (i.e., the mean coefficient of determination ($r^2$) between helping and relatedness was 0.31).

**Prospects of territory inheritance and helping behaviour.** Despite a clear overall pattern for helpers in species with territory shortage to help more and be less likely to preferentially provision close kin, there is still substantial unexplained variation in the strength of kin discrimination across species

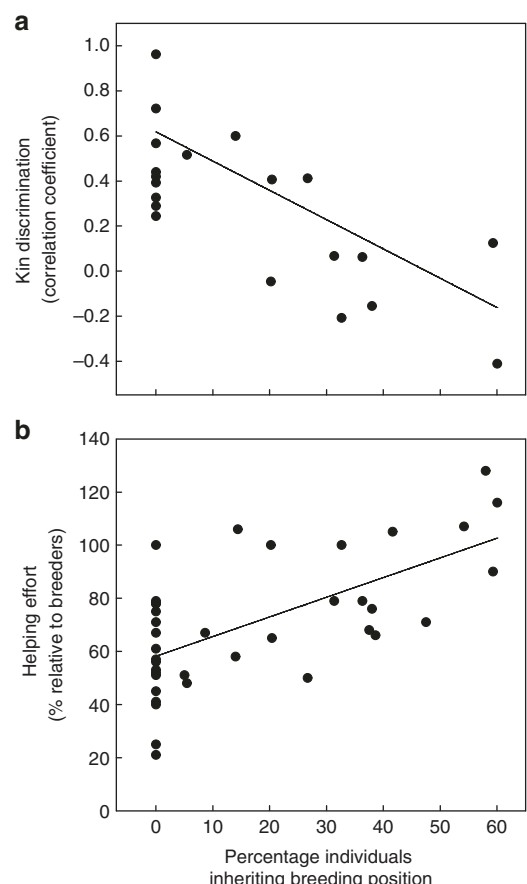

**Fig. 2** The likelihood of territory inheritance drives helping decisions in cooperatively breeding birds. **a** In cooperative breeding bird species, helpers with a high likelihood of inheriting their resident territory do not invest more in more related offspring (low levels of kin discrimination), whereas when prospects of territory inheritance are limited, subordinates mainly direct help towards related offspring (PGLS model: $n = 20$ species, $P = 0.0001$; model output is provided in Supplementary Table 3). **b** Therefore, helpers provision offspring on average more (mean % offspring food provisioning per helper, relative to breeders) when the probability of inheriting their resident territory is larger (PGLS model: $n = 38$ species, $P < 0.0001$; model output is provided in Supplementary Table 4). Dots reflect species averages, and model-predicted regression lines are plotted

(see Supplementary Table 1). To test the prediction that this variation is associated with interspecific differences in prospects of territory inheritance, I conducted a second set of analyses based on interspecific variation in the probability of territory inheritance (ranging from 0 to 60%; Supplementary Data 1). In line with the previous analysis, I found that territory inheritance explains 41% of the variation in helping behaviour across species. Both kin discrimination (PGLS: $t = -5.046$, $n = 20$ species, $P = 0.0001$; Supplementary Table 3) and helping effort (PGLS: $t = 4.515$, $n = 38$ species, $P < 0.0001$; Supplementary Table 4) are highly correlated with the probability of territory inheritance (Fig. 2) (as for the analyses with "territory shortage", the latter results were similar when "non-helping" subordinates were excluded from the calculation of helping effort; Supplementary Table 4b). Moreover, the results were similar when rates of extra-pair paternity were corrected for (Supplementary Table 4c) and when only species with a shortage of territories were included (Supplementary Table 4d).

## Discussion

The combined results of this study strongly suggest that variation in prospects of territory inheritance explains why helping effort is so variable, and why helpers preferentially direct care to related individuals in some, but not in other, cooperatively breeding species.

Why do subordinates invest in feeding unrelated individuals if they live in a territory where they may reproduce in the future? Several complementary mechanisms are probably involved, based on benefits of group living that operate immediately or in the future, alongside or in place of kin selection[21]. Individuals may be more likely to stay and willing to "pay" more in helping if the benefits of staying are larger (pay-to-stay[20]). As such, it can be predicted that individuals are selected to help more if they can inherit the territory in the future. The result that increasing prospects of territory inheritance lead, irrespective of kin-selected benefits, to higher helping effort seems to suggest that this is indeed the case. However, if higher inheritance rates are the result of the lack of options for subordinates to survive independently outside their resident territory, it could also be that breeders in highly saturated habitat can afford to force subordinates to help more because subordinates are not able to leave successfully (as predicted by biological market theory[19] and related to reproductive skew models[22]). If individuals indeed pay more to stay if outside options are limited, higher territory inheritance rates per se would not necessarily be the cause of higher helping effort but both would rather be the effect of the lack of outside options. Since only a few studies have tested the pay-to-stay hypothesis in cooperatively breeding birds and these have produced mixed results[14,20,23,24], it would be worthwhile to test whether subordinates are indeed forced to pay more when constraints for independent breeding are more severe.

In addition to, or regardless of pay-to-stay motivations for helping, if helping leads to larger groups (as is the case in many cooperative breeders[25]) a high prospect of territory inheritance itself will also promote helping behaviour for a number of reasons[21]. Larger groups are more stable and/or better able to defend the territory in many species[26], and helping to improve the group therefore facilitates territory persistence and improves the chance of individuals inheriting the territory. Moreover, larger groups may expand the territory so that subordinates can split off a part of it[27], a common route to independent breeding in some species (e.g., laughing kookaburras *Dacelo novaeguineae*[28], Florida scrub-jays *Aphelocoma coerulescens*[29]). Additionally, helping as a subordinate may lead to improved future breeding success after becoming a breeder in the territory because the resulting augmented group contains future helpers[30]. Thus, benefits of group augmentation may well explain why helpers help more if they have higher prospects of inheriting the territory in the future.

Regardless of the mechanism, if individuals can inherit the territory and queues for inheritance are stable (as is usually the case[21]), mutualistic and reciprocal benefits provided by newly recruited group members can maintain cooperation in a self-reinforcing way. This is because recruits raised by a helper will in turn help the now-breeder in order to pay to stay, to improve their own chances of inheriting the territory in the future, or to obtain benefits of group living[15,21]. This offers an adaptive explanation for why both related and unrelated individuals help substantially in bird species where territory inheritance is common, and presumably also in cooperative species in other taxa in which subordinates can inherit their resident territory, including mammals (e.g., dwarf mongoose *Helogale parvula*[31]), fish (e.g., *Neolamprologus pulcher*[32]), and insects (e.g., paper wasps *Polistes dominulus*[33]). Moreover, such

direct benefits likely also explain why in many species individuals allow unrelated immigrants to join their group[26] or even kidnap offspring from neighbouring groups (as in white-winged choughs *Corcorax melanorhamphos*[34] and pied babblers *Turdoides bicolor*[26]).

The finding that direct benefits of philopatry and territory inheritance can predict helping behaviour where kin selection cannot has substantial implications for our understanding of helping behaviour, group living, and cooperation in general. The idea that altruism can be maintained by mutualism and/or reciprocity is already firmly incorporated in "broad cooperation theory"[5,12,21,35]—but compelling comparative evidence in the context of cooperative breeding has been missing so far. While helpers clearly discriminate based on kinship in some species (see ref. [7]), direct fitness benefits appear equally, if not more, important in explaining helping behaviour in many others (e.g., refs. [14,29,36]). As such, this study provides evidence for an alternative to the prevailing paradigm that kin selection drives the evolution of helping behaviour and cooperative breeding.

## Methods

**Data collection**. Data were collected on all cooperatively breeding bird species by searching Web of Science (keywords: "Cooperative* breeding" on 30 and 31 November 2016), species-account books on cooperatively breeding birds[24,37], and by forward and backward searching citing and cited articles. Species where >10% of breeding attempts involve multiple same-sex individuals producing offspring in one nest (i.e., polygynous, polyandrous, and polygynandrous species with joint-nesting or coalitions of males or pairs[38,39]) were not included in this study because "helping" in such species could be driven by the acquisition of own parentage (e.g., Karoo scrub-robin *Erythropygia coryphaeus*[40], dunnock *Prunella modularis*[41], ground tit *Pseudopodoces humilis*[42], chestnut-crowned babbler *Pomatostomus ruficeps*[43], brown jay *Psilorhinus morio*[44], and Guira cuckoo *Guira guira*[45]). For three species with occasional joint nesting by females (moorhen *Gallinula chloropus*[46], purple gallinule *Porphyrio martinica*[47], and Seychelles warblers *Acrocephalus sechellensis*[48]), only reported data on non-parent helpers were used. Thus, the data set includes only species in which a breeding pair (breeders) is assisted by usually non-breeding subordinate helpers. The full data set is provided in Supplementary Data 1.

**Social system and territory inheritance rates**. The social system of cooperative breeding birds is different across species[16]. In order to test the prediction that territory inheritance is an important driver for helping behaviour, I used a dichotomy of whether or not independent breeding was constrained by a shortage of opportunities (i.e., territories) for independent breeding. In many cooperative breeders, subordinates are retained individuals who delayed dispersal and reproduction, and remained in a territory as a consequence of habitat saturation (a shortage of vacant breeding territories due to a lack of suitable breeding habitat, including plural breeding species in which multiple females may reproduce in independent nests in a territory). In other species, group living and helping is not the consequence of a shortage of territories. These species are: (i) non-territorial because individuals breed in colonies and nesting space is not limited (e.g., pied kingfishers *Ceryle rudis*[49] and sociable weavers *Philetairus socius*[50]), (ii) species with "redirected care" in which individuals help others after failing to attract a breeding partner (e.g., pygmy nuthatch *Sitta pygmaea*[51]) or after their own breeding attempt failed (e.g., long-tailed tit *Aegithalos caudatus*[52] and rifleman *Acanthisitta chloris*[53]), or (iii) species in which helpers are sexually immature juveniles (moorhen[54] and purple gallinule[47]). I classified each species' social system (retained offspring, plural breeding, colonial, redirected care, or immature juvenile helpers) based on the description of their social system in the original publications reporting the collected data (see Supplementary Data 1 for an overview). These original publications invariably state the origin of subordinates (retained offspring, failed breeders from elsewhere or immature individuals) and describe whether the study species breeds in colonies or not (see Supplementary Data 1 for references).

For the 16 species without shortage of territories, inheritance levels were set to zero because helpers had no possibility to inherit a territory. For 22 of the 28 species with shortage of territories, I was able to obtain data on the proportion of subordinates that eventually inherited all, or part, of their resident territory (i.e., subordinates eventually reproduced in the territory where they helped; Supplementary Data 1). In three species in which helpers were retained offspring, a small proportion of helpers had attempted to breed independently earlier in the season before becoming helpers (i.e., redirected care, see (ii) above), but these species were considered a species with shortage of territories because the vast majority of helpers were staying and helping in their resident territory due to habitat constraints (Rufous treecreeper *Climacteris rufa*[55]; brown treecreeper *Climacteris picumnus*[56]; Galápagos mockingbird *Mimus parvulus*[57]).

**Helping effort**. In order to quantify helping effort, I followed procedures outlined in Green et al[10]. Briefly, since absolute food provisioning rates are not comparable between species, I calculated the percentage of food provisioning trips made by subordinates in relation to that of breeders of the same sex. Since breeders may adjust their investment based on whether they have helpers or not[25], only breeders that did have helpers were used as reference group[10]. In contrast to Green et al.[10], I (conservatively) included non-helping subordinates in the calculation of helping effort since (a) non-helping subordinates may inherit their resident territory and (b) non-helpers were included in measures of average helping effort in some studies and it was not known what percentage of subordinates did not help (red-winged fairy-wren *Malurus elegans*[58] and Australian magpie *Gymnorhina tibicen*[59]). I repeated the analyses excluding non-helpers in the calculation of helping effort, but this did not change the results (see below). Moreover, where Green et al.[10] used specific subsets of data for three species, I included the whole available data set for the calculation of helper effort: for El Oro parakeets *Pyrrhura orcesi*[60] and purple gallinules[47] I included yearlings as well as older helpers, and I included individuals that helped in their natal territory as well as immigrated helpers for pied kingfishers[49].

**Kin discrimination**. The strength of kin discrimination was defined as the extent to which subordinates preferentially direct care towards related offspring, and for each species calculated as the effect of relatedness between helpers and beneficiaries (correlation coefficient, $r$) on the probability of helping or the amount of provided help (following Griffin et al.[7] and Cornwallis et al.[8]). Values of kin discrimination were obtained from Cornwallis et al.[8], or, for studies published since, calculated based on formulae provided in Lajeunesse et al.[61] for transforming common statistical metrics (e.g., $t$, $F$, $\chi^2$) into a correlation coefficient. For species with multiple estimates of kin discrimination (see Supplementary Table 1), I used the average correlation weighed by sample size. The value for kin discrimination in green wood hoopoes (*Phoeniculus purpureus*) provided in Cornwallis et al.[8] was not included, because the experiment on kin discrimination was based on an unnatural situation (addition of two nest boxes, one with related and one with unrelated nestlings, close to the place of the removed original nest box[62]).

**Other group and helper characteristics**. Data on average group size, relatedness between helpers and beneficiaries, subordinate sex ratio, and levels of extra-pair paternity were obtained preferably from the same sources or population from which data on helping effort, kin discrimination, and territory inheritance rates were obtained, and included helpers as well as non-helping subordinates. The methods used to estimate relatedness vary between studies (being either based on genealogical data or molecular genetic data), and although this was shown to not substantially affect relatedness estimates[9], data were collected based on genetic estimates where possible[10].

**Statistical analyses**. For all analyses, PGLS models were constructed using the caper package[63] in R 3.3.0[64]. I applied a maximum likelihood estimation of Pagel's $\lambda$ for phylogenetic dependence[65,66], although the correlations showed very limited, phylogenetic structure (i.e., estimates of $\lambda$ were smaller than 0.001 in all analyses). Uncertainty in phylogenetic relationships between species was accounted for by repeating each model using 1000 phylogenetic trees. These trees (using the Hacket et al.[67] backbone) are based on a recent comprehensive phylogenetic avian phylogeny[68], and were obtained from http://birdtree.org. For each test, I report the mean estimates and two-tailed significance values of these 1000 models including all explanatory variables. Model assumptions of normality and homoscedasticity were confirmed visually (by respectively plotting model predictions against residuals, and inspecting the distribution of residuals using histograms and Q–Q plots).

To test the prediction that increased probability of territory inheritance should reduce selection on subordinates to discriminate based on their relationship to offspring they (might) provision, I assessed whether shortage of territories for independent breeding (yes/no) and territory inheritance rate predict the degree of kin discrimination across species (Supplementary Tables 1 and 3). As additional predictors, I included the method used in each study to measure kin discrimination (probability of help or amount of help; or the method with highest sample size if both were used) as well as the average relatedness of subordinates (since this may determine the potential for kin discrimination in the first place), but none of these variables had a significant effect on kin discrimination (Supplementary Tables 1 and 3).

To test the prediction that shortage of territories for independent breeding (habitat saturation) and the prospects of territory inheritance are important drivers of variation in helping behaviour, I assessed whether helping effort (response variable) was predicted by (1) a shortage of territories for independent breeding (Supplementary Table 2a) or by (2) the average probability of territory inheritance (Supplementary Table 4a) in two separate models. The analysis of the effect of territory shortage contained two species (red-backed fairy-wren and American crow *Corvus brachyrhynchos*) with exceptionally high helping effort (Supplementary Data 1). I therefore log10-transformed values of helping effort in this analysis in order to avoid a large right skew of data distribution. These two species were not present in the analysis of territory inheritance (because probability of territory inheritance was unknown for both), so helping effort data were not transformed in that analysis. As helping effort may be affected by the average relatedness between subordinates and offspring, average group size (log10 transformed) and helper sex ratio[10], I included these data as additional covariates in both models (but none of these variables had a significant effect on helping effort; Supplementary Tables 2 and 4). I repeated these two models, excluding non-helping subordinates in the calculation of helping effort for 12 species (see Supplemental Data 1), but the results were similar (Supplementary Tables 2b and 4b; two species were not included in these analyses since it was not known what percentage of subordinates did not help; see above). I ran two additional models based on potential confounding variables that were not available for all species. (i) Values of helping effort may be inflated if breeder males (included in the reference group to define helper effort; see above) reduce their effort as a result of extra-pair mating by their social partner. Therefore, I repeated the models with helping effort as the response variable, including the above-mentioned predictors and rates of extra-pair paternity (percentage broods with at least one extra-pair offspring) as explanatory variables. I used a subset of species for which rates of extra-pair paternity were available ($n = 31$ of 44 species and 26 of 38 species for models with territory shortage; Supplementary Table 2c, and territory inheritance rate, Supplementary Table 4c, as explanatory variables, respectively). (ii) I also assessed whether the effect of the average probability of territory inheritance on helping effort was similar if only species that lived in saturated habitats were included (Supplementary Table 4d). Despite the lower sample sizes in these tests, the results of these additional subset models were similar to the results from the models including all data (Supplementary Tables 2 and 4).

To determine whether less kin discrimination leads to higher average helping effort (as predicted if unrelated subordinates also help at full capacity), I tested whether these two variables were correlated for the 21 species for which both variables were available.

Comparative analyses require data of sufficient quality, collected in a consistent way. However, some data included in the current study may be interpreted in a non-objective way (see below). Therefore, I also ran conservative analyses excluding species with potentially ambiguous data (highlighted in Supplementary Data 1). For the conservative analyses of helping effort, I excluded white-browed sparrow weavers *Plocepasser mahali* (provisioning rates for helpers and breeders were obtained in different populations[69]), pied babblers *Turdoides bicolor* (helping effort was based on percentage food given up; not provisioning per se[70]), white-winged choughs *Corcorax melanorhamphos* (just one group was included[10]), and Australian magpies (unclear if breeders without helpers were included in breeder provisioning rate[59]). For the conservative analyses of territory inheritance rates I excluded the species for which inheritance rates were unclear or based on very small sample sizes (Rufous vanga *Schetba rufa*, purplish-backed jay *Cyanocorax beecheii*, toucan barbet *Semnornis ramphastinus*, El Oro parakeet, pygmy nuthatch *Sitta pygmaea*; Supplementary Data 1). The results of these conservative analyses are, however, similar to the results including all species (Supplementary Table 5).

**Code availability**. The code used for statistical analyses is available upon request.

**Data availability**. The complete data set containing data on 44 cooperatively breeding species is available in Supplementary Data 1.

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

## Acknowledgements

I thank K. Bebbington and K. Delhey for comments. The research was funded by the Netherlands Organisation for Scientific Research (NWO; VENI-fellowship 863.13.017).

## Author contributions

S.A.K. designed the study, collected the data, conducted the analyses, and wrote the paper.

## Additional information

**Competing interests:** The author declares no competing financial interests.

