## [Peer Review File · Nature Communications]

Reviewers' comments:

Reviewer #1 (Remarks to the Author):

Beyond kin selection: direct benefits explain interspecific variation in helping behavior

This ms is an interesting consideration of kin selection versus direct benefits as an influence on helping rates in cooperative breeders. However, some aspects of the analysis are quite unclear. Detailed comments are as follows

L 57: this makes the implicit assumption that if territory inheritance is not the reason for cooperation, then kin selection is. There could be other direct benefit reasons that explain cooperation, not necessarily a reversion to kin selection

L 66: how are levels of helping behavior compared across species where researchers have taken different measures of level of help?

L 75-80: I don't see what kin discrimination has to do with territory inheritance: species could be able to discriminate kin for various reasons that are not related to helping behaviour (such as inbreeding avoidance)

The author/s state that relatedness only explain 10% of variation in helping behavior, but do not give a % of variation in helping behavior that is explained by territory inheritance

L 94-95: note that in cooperative breeders there is likely to be an upper group size limit, beyond which the benefits of larger groups decline. This critical group size effect, or social queue effect, has already been identified in a number of species.

I have noted a couple of places where the references given do not align with the numbers assigned to them in text – please check this discrepancy.

Figure 1 & 2(b): is this average helping effort, or cumulative? Differences in average group sizes between species will affect this figure, so would need to be averaged over helpers.

L 244-5: this description does not provide enough explanation as to how habitat saturation was inferred from published data – 'based on published descriptions of the breeding system and the origin of subordinates'. Further down in the analysis section it is clear that how habitat saturation was measured is a very important consideration, given the analyses conducted. I would therefore like to see greater explanation of the calculation.

L 273-277: to me, these reasons do not suffice to include non-helping subordinates. The key question here is why do subordinates help and why do they show different levels of help, so including species in which subordinates don't help would seem to skew the data here.

L 289: how was his comparable between species? IN terms of (a) amount of help, and (b)

did you use probability of helping for some species, and amount of help for others? If yea, how was this comparable?

L 328: typo

L 340: why was this two separate models? Could these two predictors not have been included in the same model for better comparison? Further down it is clear that other predictors we included on the models, so why not the two key parameters

I am not convinced that the term kin discrimination is the best term to describe the question being asked here. Kin discrimination is often used to test whether individuals recognize relatives of varying relatedness using discrimination and ID tests

In the supplementary dataset, the sample sizes seem fairly low, especially considering some were excluded from some analyses. This leads me to wonder whether the models were overparameterized and the results reliable. Indeed, when looking at the extended data tables, I see five predictors for a sample size of 22, which seems too many predictors relative to sample size to generate reliable estimates.

Reviewer #2 (Remarks to the Author):

The importance of direct versus indirect benefits in driving the evolution of cooperative breeding behavior is an important but debated topic. This study takes a novel, comparative approach to test between these hypotheses in birds. The author makes a series of predictions about how the prospect of territory inheritance might influence why subordinates not only remain in the group, but help unrelated individuals. The author concludes that subordinates provide more help in species where the prospects of territorial inheritance or higher. They take this as evidence of direct benefits as being important in the evolution of cooperative breeding behavior.

This is an intriguing result and the paper is clear, well-written, and very compelling. I only have two concerns, which I outline in greater detail below. First, I have some questions about the decisions to exclude certain species and not others. Second, and perhaps more important, I would like to see this work but into the classic ecological constraints and habitat saturation framework more. Territorial inheritance is likely to occur more in species where habitat is saturated and subordinates have few outside options to disperse and breed on their own because there is no place to go. Reproductive skew theory has long dealt with this issue of helping effort and territoriality, yet there is no mention of this in the paper. It seems that the results presented here are consistent with some of the flavors of reproductive models. It would be important to discuss these results in that context and then explain why thinking about direct benefits in this way is a better way to explain the data.

L56. Territoriality versus non-territoriality is really just territorial versus colonial and redirected helping grouped together in your data, which are VERY different phenomena.

Rather than lumping them together, I think you should break them out. There is very strong evidence that redirected helping is driven by kin benefits in species like long-tailed kits and others. I realize your sample size is small, but I worry that you are loading the deck a bit by sticking them together. Could redirected helpers be driving this result (in Fig 1A)?

L63. There are alternative models—things like reproductive skew theory—which are based on the idea of kin selection but incorporate habitat saturation. They make predictions about helping effort (e.g. Figure 1B) in relation to habitat constraints. Your results for this analysis seem to be consistent with some of these models. For example, helping effort will be higher in species where independent breeding is more constrained, and not necessarily because of the potential for territory inheritance. In other words, can you explain this result in the absence of territorial inheritance being the driving force with something that may be more parsimonious? I realize that Figure 2B at least partially ameliorates this concern, but are the prospects of territorial inheritance higher because the constraints on independent breeding are so much higher? In other words, are the observed relationships ultimately related to habitat saturation, delayed dispersal, etc. and only secondarily to territorial inheritance.

L223. I have a few questions about your choice of species to include or exclude. In your definition, you are effectively limiting your dataset to primarily singular breeders and excluding many plural breeding species (brown jay, chestnut-crowned babbler) and joint-nesting species. While I don't take issues with this, many of the species you did decide to include in the dataset are plural breeders and perhaps seem to meet these criteria (bell minor, Galapagos mockingbird, sociable weaver, white-winged chough, etc), especially if you avaried that >10% threshold.

I think doing some analyses to look at these factors (e.g., plural versus singular breeding) with the species you included in your dataset will be useful and informative. If subordinates have options to breed within the group, then that might also explain helping non-relatives beyond territorial inheritance. In fact, this seems like a very plausible alternative hypothesis that you could test with your data.

L269. In many species, non-breeding subordinates perform other important roles like nest defense. What happens to your analysis of helping behavior if you remove these individuals and only consider individuals that actually bring food to the nest?

Appendix. It would help to make sure the variables in the models/tables have the same names in the appendix. For example, "habitat saturation" should be the same as "territorial shortage", as should "rate of extra-pair paternity" and "% nests with extra-pair young", etc. By using different names in your appendix, it becomes harder to figure out which data are being used in which models.

Revision NCOMMS-17-10999-T "Direct benefits explain interspecific variation in helping behaviour among cooperatively breeding birds"

Dear Reviewers,

Thank you very much for the constructive and very thoughtful comments and suggestions regarding the manuscript "Direct benefits explain interspecific variation in helping behaviour among cooperatively breeding birds". Here, I resubmit a substantially revised manuscript with incorporated changes based on your comments and suggestions. I think that the amendments, changes and additional analyses have considerably improved the manuscript, especially since they facilitated a broadening of the message and increased clarity in the methodological approaches. I am therefore very grateful for the comments.

Please find below my detailed responses and highlighted changes in bold font. I have numbered the comments to enable easy cross-referencing and added how and where in the manuscript changes were made (cited text is given in italic font). Line numbers refer to line numbers in the revised manuscript unless indicated otherwise.

A few general remarks concerning recurring comments are perhaps worth mentioning here:

1. A number of comments derived from missing definitions and explanations of data selection methodology in the main text of the original submission. I realised now that, because this important information was somewhat hidden in the Methods and Supplemental Tables sections, some statements in the main text may have been difficult to interpret. I have now provided the relevant details in the main text, so that the reviewers' questions are answered early in the manuscript. For example, I now describe that the data are comparable across species (comments 3, 11), define 'kin discrimination' (Comments 4, 14), explain that helping effort was calculated as average per helper (Comment 8) and explain in more detail how 'habitat saturation/shortage of territories' was defined for each species (Comment 9). Please see specific responses to the relevant numbered comments for details.
2. The reviewers ask for an explanation for why certain criteria were used for some analyses and also requested some additional analyses (comments 5, 9, 10, 11, 13, 17, 19, 20). For example, both reviewers asked how certain methodological changes would affect the results (e.g. comments 10 and 20). These comments are valid and very useful, and I have now explained the criteria in more detail and provided the requested analyses. I would like to mention here that I used conservative analyses throughout the manuscript and that I originally explored potential confounds and alternative hypotheses provided in the Supplemental Tables; I have now referred to these analyses in the Results section of the main article.
3. The second reviewer's comments 17 and 19 suggest the potential further splitting up of the categories 'with and without territory shortage' into more distinct classes of species' social system (e.g. species with redirected care vs. colonial species; comment 17). As the reviewer rightly points out, the overall result could potentially be driven by an effect in species with a particular social system (thereby potentially masking a lack of effect in others). Unfortunately, the limited sample size for species with each social system does not currently allow for appropriate statistical testing. However, I have now reported mean values for each social system in the result section in main text. These data suggest that the overall reported patterns are consistent across different social systems within each category (e.g. both in species with

redirected care and colonial species, kin discrimination is high and helping effort low). I think that this convincingly shows readers that the results are not selectively driven by species with different social systems.

With many thanks for your consideration.
I look forward to hearing from you again.

Yours sincerely

Reviewer #1 (Remarks to the Author):

Beyond kin selection: direct benefits explain interspecific variation in helping behavior

Comment 1. This ms is an interesting consideration of kin selection versus direct benefits as an influence on helping rates in cooperative breeders. However, some aspects of the analysis are quite unclear. Detailed comments are as follows

Response: Thank you very much for your insightful comments and suggestions. I agree that aspects of some analyses were unclear and I think this is mainly because these should have been added or explained in more detail in the main text (rather than only in the Methods and Extended Data sections). I am very grateful for this advice. I have added the relevant details and additional data in the main text (see my responses to your comments below for details), which I believe has clarified the logic and analyses substantially.

Comment 2. L 57: this makes the implicit assumption that if territory inheritance is not the reason for cooperation, then kin selection is. There could be other direct benefit reasons that explain cooperation, not necessarily a reversion to kin selection

Response: This sentence indeed implied an unnecessary dichotomy between kin selection and territory inheritance (L. 54-57 in the original manuscript: *“Specifically, if all territories in the population are occupied and helping promotes territory inheritance, subordinates should help at full capacity regardless of kinship, whereas if territories are not limited, help should predominantly be directed to highly related individuals”*). I have now removed the second part of the sentence and toned down the statement that individuals should help at full capacity, in order to make clear that if other benefits than kin-selected benefits are important, we would not expect a correlation between helping effort and relatedness. L. 58-60 now reads: *‘Specifically, if all territories in the population are occupied and helping promotes territory inheritance, subordinates should help regardless of whether they are related to the recipient of their help.’*

Comment 3. L 66: how are levels of helping behavior compared across species where researchers have taken different measures of level of help?

Response: To quantify helping behaviour I used offspring food provisioning rates for all species, since these data are widely available and easily quantified, and because they are comparable across species suitable for comparative studies (as was also done in e.g. Green et al. 2016 Nature Comm.). Based on this comment, I now realised that nowhere in the main text had I actually defined ‘helping behaviour’ as such, and I have now added this in the following sections:

L. 8-12: *‘Here, I use comparative analyses to show that direct fitness benefits of helping others, associated with future opportunities to breed in the resident territory, are responsible for the widespread variation in helping effort (offspring food provisioning) and kin discrimination across cooperatively breeding birds.’*

L. 60-64: *‘To test this hypothesis, I collected data on helping effort (measured as average offspring provisioning rates of subordinates relative to breeders of the same sex¹⁰) and degree of kin discrimination (the species-specific correlation coefficient between relatedness and helping^{7,8}) from published papers on 44 cooperatively breeding species.’*

L. 81-84: *‘in species with territory shortage subordinates invest on average 51% more in offspring provisioning than subordinates in species where territory shortage does not constrain independent breeding.’*

L. 141-142: 'Why do subordinates invest in feeding unrelated individuals if they live in a territory where they may reproduce in the future?'

Since comparative studies rely on comparable data, I also want to highlight here that I conducted a conservative analysis in which I removed species for which provisioning data had been collected in a potentially non-comparable way (Extended Data Table 5). This analysis yielded similar results.

L. 359-365: 'For the conservative analyses of helping effort, I excluded white-browed sparrow weavers *Plocepasser mahali* (provisioning rates for helpers and breeders were obtained in different populations^{67,68}), pied babblers *Turdoides bicolor* (helping effort was based on percentage food given up; not provisioning per se⁶⁹), white-winged choughs *Corcorax melanorhamphos* (just one group was included¹⁰) and Australian magpies (unclear if breeders without helpers were included in breeder provisioning rate⁵⁷).'

Comment 4. L 75-80: I don't see what kin discrimination has to do with territory inheritance: species could be able to discriminate kin for various reasons that are not related to helping behaviour (such as inbreeding avoidance)

Response: One aim of this study is to determine why unrelated individuals help, since kin-selection cannot explain this behaviour. The usage of 'kin discrimination' (the species-specific correlation coefficient between relatedness and helping behaviour) as a comparable measure for whether care is mainly directed to related individuals or not, is an extremely useful metric for this purpose (see e.g. refs 7-9: Griffin & West 2003 Science; Cornwallis et al. 2009 J. Evol. Biol; Cornwallis et al. 2010 Nature). I realise, however, that this was indeed not clear and have now 1) added the definition of kin discrimination, 2) made the reasoning for this clearer and 3) explained how this was calculated, while referring to the original studies using this approach:

- L. 32-36: '*To understand the evolutionary maintenance of cooperative breeding, and cooperation and sociality more generally, we must determine within and across species: (i) the factors responsible for the widespread variation in helping behaviour and (ii) the extent to which help is preferentially directed to more related individuals^{5,12}.*'

- L. 60-63 '*...I collected data on helping effort (measured as offspring provisioning rates of subordinates relative to breeders of the same sex¹⁰) and degree of kin discrimination (the species-specific correlation coefficient between relatedness and helping^{7,8}).*'

- L. 66-79: '*The analyses revealed that prospects of territory inheritance are responsible for a large part of the currently unexplained variation in helping behaviour and kin discrimination (i.e. the extent to which helping behaviour is preferentially directed to more related individuals^{7,8}) in cooperatively breeding birds.*'

I appreciate that, in general, the term kin discrimination can be used for other behaviours also, but I hope that its usage in this article has become clear with addition of the current definition. The term is commonly used for describing the extent to which helping behaviour is preferentially directed to more related individual (see references above).

Comment 5. The author/s state that relatedness only explain 10% of variation in helping behavior, but do not give a % of variation in helping behavior that is explained by territory inheritance

Response: This is a good point and important to mention. I have now added that across species 41% of the variation in helping effort is explained by variation in probability of inheritance in L. 127-128: '*I found that territory inheritance explains 41% of the variation in helping behaviour across species.*'

Comment 6. L 94-95: note that in cooperative breeders there is likely to be an upper group size limit, beyond which the benefits of larger groups decline. This critical group size effect, or social queue effect, has already been identified in a number of species.

Response: I agree that helping behaviour in large groups might not necessarily lead to larger groups. In this section I aim to explain the benefits that subordinates may obtain from helping should they have the opportunity to inherit the territory, providing three non-exclusive explanations (as introduced using the question *'Why do subordinates invest in feeding unrelated individuals if they live in a territory where they may reproduce in the future?'* in L. 141-142). These benefits are not restricted to those associated with group augmentation (help leading to larger groups), but also with pay to stay (as explained in L. 142-170). I have now rewritten this part so that it becomes clear that if helping leads to larger groups, this is just one of the potential explanations for why subordinates help more in species that are more likely to inherit the territory: l. 57-160: *'However, in addition to, or regardless of pay-to-stay motivations for helping, if helping leads to larger groups (as is the case in many cooperative breeders²³) a high prospect of territory inheritance itself will also promote helping behaviour for a number of potential reasons²¹.'*

Comment 7. I have noted a couple of places where the references given do not align with the numbers assigned to them in text – please check this discrepancy.

Response: Thank you for pointing this out – I have corrected this throughout the manuscript and the Supplemental table.

Comment 8. Figure 1 & 2(b): is this average helping effort, or cumulative? Differences in average group sizes between species will affect this figure, so would need to be averaged over helpers.

Response: Helping effort was indeed expressed as *average effort per helper*. I have now made this clear in the figure legends (L. 847 and 862: *'helping effort (average % offspring food provisioning per helper, relative to breeders)'*).

And in the main text (L. 60-62): *'To test this hypothesis, I collected data on helping effort (measured as average offspring provisioning rates of subordinates relative to breeders of the same sex¹⁰)'*

Comment 9. L 244-5: this description does not provide enough explanation as to how habitat saturation was inferred from published data – 'based on published descriptions of the breeding system and the origin of subordinates'. Further down in the analysis section it is clear that how habitat saturation was measured is a very important consideration, given the analyses conducted. I would therefore like to see greater explanation of the calculation.

Response: Thank you for this remark; I agree that this definition is important and needs to be explained carefully. I have now rewritten this section (headed: *'Social system, shortage of territories and territory inheritance rates'*; see below, especially the underlined part) to make clearer how 'shortage of territories' was inferred (authors of each included study clearly outlined and described the social system) and how the social system was defined. I realised that in the methods section I also used 'habitat saturation' rather than 'shortage of territories' and have also changed that throughout the methods section.

L. 211-243:

'Social system, shortage of territories and territory inheritance rates

*The social system of cooperative breeding birds is different across species¹⁶. In order to test the prediction that territory inheritance is an important driver for helping behaviour, I used a dichotomy of whether or not independent breeding was constrained by a shortage of opportunities (i.e. territories) for independent breeding. In many cooperative breeders, subordinates are retained individuals who delayed dispersal and reproduction, and remained in a territory as a consequence of habitat saturation (a shortage of vacant breeding territories due to a lack of suitable breeding habitat; including plural breeding species in which multiple females may reproduce in independent nests in a territory). In other species, group living and helping is not the consequence of a shortage of territories. These species are either: (i) non-territorial because individuals breed in colonies and nesting space is not limited (e.g. pied kingfishers *Ceryle rudis*⁴⁷ and sociable weavers *Philetairus socius*⁴⁸), (ii) species with 'redirected care' in which individuals help others after failing to attract a breeding partner e.g. pygmy nuthatch *Sitta pygmaea*⁴⁹ or after their own breeding attempt failed (e.g. long-tailed tit *Aegithalos caudatus*⁵⁰ and rifleman *Acanthisitta chloris*⁵¹), or (iii) species in which helpers are sexually immature juveniles (moorhen⁵² and purple gallinule⁴⁵). I classified each species' social system (retained offspring, plural breeding, colonial, redirected care or immature juvenile helpers) based on the description of their social system in the original publications reporting the collected data (see Supplementary Data 1 for an overview). These original publications invariably state the origin of subordinates (retained offspring, failed breeders from elsewhere or immature individuals) and describe whether species breeds in colonies or not (see Supplementary Data 1 for references).*

*For the 16 species without shortage of territories, inheritance levels were set to zero because helpers had no possibility to inherit a territory. For 22 of the 28 the species with shortage of territories, I was able to obtain data on the proportion of subordinates that eventually inherited all, or part, of their resident territory (i.e., subordinates eventually reproduced in the territory where they helped; Supplementary Data 1). In three species in which helpers were retained offspring, a small proportion of helpers had attempted to breed independently earlier in the season before becoming helpers (i.e. redirected care, see (ii) above), but these species were considered a species with shortage of territories because the vast majority of helpers were staying and helping in their resident territory due to habitat constraints (rufous treecreeper *Climacteris rufa*⁵³; brown treecreeper *Climacteris picumnus*⁵⁴; Galápagos mockingbird *Mimus parvulus*⁵⁵).*

Comment 10. L 273-277: to me, these reasons do not suffice to include non-helping subordinates. The key question here is why do subordinates help and why do they show different levels of help, so including species in which subordinates don't help would seem to skew the data here.

Response: I included non-helping subordinates for three reasons: (1) non-helpers may still inherit the territory, (2) helping effort could not be calculated excluding non-helping subordinates for all species (because the number of non-helping subordinates was not given for some species), and (3) these measures are conservative for the analyses. In species with territory shortage, some subordinates do not help and thus drive down the average helping effort – this goes against the hypothesis that helping effort should be high in such species. In species without territory shortage, there are no non-helpers included in the calculation of helping effort (because there are no non-helping subordinates in a group), which goes against the hypothesis that helping effort should be low in such species.

Nonetheless, as per the reviewer's request, I have now added analyses where non-helping subordinates are excluded in the calculation of helping effort. The repeated analyses give similar

results (see Supplementary Table 2a vs 2b, and 4a vs. 4b), but the effect sizes for the variables 'territory shortage' and 'percentage inheritance' appear slightly larger (confirming the conservative nature of the initial analyses). Moreover, I have now added to the text that the main data did include non-helpers but that the results did not change when excluding non-helping subordinates:

L. 84-88: *'In 12 species some subordinates did not contribute to feeding offspring (Supplementary Data 1) and these were included in the calculation of helping effort. The results are, however, similar when these 'non-helping' subordinates are excluded from the calculation of helping effort (Supplementary Table 2b).'*

L. 131-133 *'similar as for the analyses with 'territory shortage', the latter results were similar when 'non-helping' subordinates were excluded from the calculation of helping effort; Supplementary Table 4b).'*

In addition, I added the details about this analysis to the methods section (L. 251-257: *'I (conservatively) included non-helping subordinates in the calculation of helping effort since (a) non-helping subordinates may inherit their resident territory and (b) non-helpers were included in measures of average helping effort in some studies and it was not known what percentage of subordinates did not help (red-winged fairy-wren *Malurus elegans*⁵⁶ and Australian magpie *Gymnorhina tibicen*⁵⁷). I repeated the analyses excluding non-helpers in the calculation of helping effort, but this did not change the results (see below).'*

Comment 11. L 289: how was his comparable between species? IN terms of (a) amount of help, and (b) did you use probability of helping for some species, and amount of help for others? If yea, how was this comparable?

Response: In the analyses of kin discrimination, I used correlation coefficients obtained from studies assessing the effect of relatedness on the probability and/or the amount of help provided (following Griffin and West 2003 Science, Cornwallis et al. 2009 J. Evol. Biol., Cornwallis et al. 2010 Nature). The estimate of kin discrimination may potentially differ between these two methods and, although this appeared not the case (see Supplementary Tables 1 and 3), I included which method each study used in the model and described that this had no effect on the extend of kin discrimination. L. 308-313: *'As additional predictors, I included the method used in each study to measure kin discrimination (probability of help or amount of help; or the method with highest sample size if both were used) as well as the average relatedness of subordinates (since this may determine the potential for kin discrimination in the first place), but none of these variables had a significant effect on kin discrimination (Supplementary Tables 1 and 3).'*

Comment 12. L 328: typo

Response: Changed 'inheritances' to 'inheritance'

Comment 13. L 340: why was this two separate models? Could these two predictors not have been included in the same model for better comparison? Further down it is clear that other predictors we included on the models, so why not the two key parameters

Response: I used two different approaches, by testing whether helping behaviour and kin discrimination are associated with (1) territory shortage / habitat saturation (yes/no) and (2) the degree of territory inheritance (in %). Since helpers in species where there is no shortage of territories cannot be helping to inherit a territory, these two measures could not be included in

the same model (i.e. the degree of inheritance in species without territory shortage is always zero and these two measures would be highly collinear). Although both analyses provide a somewhat similar message, I think it is important to provide them both for two reasons: (1) I think that it is important to realise and inform readers for future work that the social systems of different cooperative breeders can be very different. (2) The actual percentage of individuals that inherit is not known for several species where the degree of territory shortage *is* known; just providing analyses of inheritance rate (i.e. Fig. 2) would reduce the sample size substantially. I think therefore that providing the analyses in Fig. 1 is a good way to start before going into more depth in Fig. 2.

Comment 14. I am not convinced that the term kin discrimination is the best term to describe the question being asked here. Kin discrimination is often used to test whether individuals recognize relatives of varying relatedness using discrimination and ID tests

Response: [See also my response to comment 4].

I agree that the term here used could be something like 'degree of kin-directed care'. Apart from that this would lead to more complex and confusing writing, the reason I have used 'kin discrimination' is that many important previous studies on this topic and in this context (including the source for some of the data) have invariably used this term (Griffin & West 2003 Science; Cornwallis et al. 2009 J. Evol. Biol; Cornwallis et al. 2010 Nature). I think it is unwise to change the semantics at this stage, as this would lead to confusion in this field. However, I have now clarified that the aim is to look at different degrees of kin discrimination in L. 304-308: '*To test the prediction that increased probability of territory inheritance should reduce selection on subordinates to discriminate based on their relationship to offspring they (might) provision....*' Moreover, I defined 'kin-discrimination' early on (l. 35-36: '*the extent to which help is preferentially directed to more related individuals*'), and I described now that I used correlation coefficients to quantify kin discrimination (L. 62-64: '*degree of kin discrimination (the species-specific correlation coefficient between relatedness and helping^{7,8}) from published papers on 44 cooperatively breeding species.*'.

Comment 15. In the supplementary dataset, the sample sizes seem fairly low, especially considering some were excluded from some analyses. This leads me to wonder whether the models were overparameterized and the results reliable. Indeed, when looking at the extended data tables, I see five predictors for a sample size of 22, which seems too many predictors relative to sample size to generate reliable estimates.

Response: I agree that some of the subset analyses have a high ratio of explanatory variables to sample size. The reason is that some additional variables were not available for all species for these analyses (e.g. levels of extra-pair paternity, proportion subordinates that help). This is why I kept the number of explanatory variables to a minimum in the main analyses with sufficiently high sample sizes (those presented in the main text, Extended Data Tables 1 and 2a, and Figure 1 and 2). In other words, it is unlikely that the analyses providing the main results suffered from overparameterization. The additional subset analyses were conducted to test if the effect of inheritance changed after controlling for other variables, without specifically being interested in the effect of those variables *per se* (of which the estimates, indeed, would perhaps be less reliable). In none of the analyses this is the case, leaving me to conclude that the main analyses were robust against alternative explanation or potential confounding factors.

Reviewer #2 (Remarks to the Author):

Comment 16. The importance of direct versus indirect benefits in driving the evolution of cooperative breeding behavior is an important but debated topic. This study takes a novel, comparative approach to test between these hypotheses in birds. The author makes a series of predictions about how the prospect of territory inheritance might influence why subordinates not only remain in the group, but help unrelated individuals. The author concludes that subordinates provide more help in species where the prospects of territorial inheritance are higher. They take this as evidence of direct benefits as being important in the evolution of cooperative breeding behavior. This is an intriguing result and the paper is clear, well-written, and very compelling. I only have two concerns, which I outline in greater detail below. First, I have some questions about the decisions to exclude certain species and not others. Second, and perhaps more important, I would like to see this work fit into the classic ecological constraints and habitat saturation framework more. Territorial inheritance is likely to occur more in species where habitat is saturated and subordinates have few outside options to disperse and breed on their own because there is no place to go. Reproductive skew theory has long dealt with this issue of helping effort and territoriality, yet there is no mention of this in the paper. It seems that the results presented here are consistent with some of the flavors of reproductive models. It would be important to discuss these results in that context and then explain why thinking about direct benefits in this way is a better way to explain the data.

Response: Thank you very much; I am very happy to read that you find the study clear, well-written, and very compelling, and that you highlight the novel approach. I am also very grateful for your insightful comments; I have considered these carefully and I think that the resulting ‘broadening’ of the manuscript’s scope has improved it substantially. I have responded to the questions “about the decisions to exclude certain species and not others” in detail below each comment (comments 17, 19, 20 and 21) and, as suggested, now discuss the results in the broader context of habitat saturation and reproductive skew theory (Comment 18).

Comment 17. L56. Territoriality versus non-territoriality is really just territorial versus colonial and redirected helping grouped together in your data, which are VERY different phenomena. Rather than lumping them together, I think you should break them out. There is very strong evidence that redirected helping is driven by kin benefits in species like long-tailed tits and others. I realize your sample size is small, but I worry that you are loading the deck a bit by sticking them together. Could redirected helpers be driving this result (in Fig 1A)?

Response: I think that it is a valid observation that colonial species and species with redirected care (and also those with immature subordinates from the same season) can be considered very different. I had initially considered splitting them up in the manuscript but the limited sample size does not permit adequate statistical testing for differences in kin discrimination and helping effort between these groups (see sample sizes in the quoted text below). This said, I completely agree that only one group could solely drive this statistical effect, and potentially mask the lack of effect in the other(s). In the absence of sample sizes appropriate for statistical analysis, I have now provided average feeding effort data and degree of kin discrimination for species with different social systems. These values strongly suggest that helping effort and kin discrimination are low for all social systems in which helpers do not inherit a territory:

L. 94-111: *For the analyses, species were categorized as species with and species without territory shortage. However, social systems vary considerably within these categories: species without territory shortage can be colonial, or involve juvenile helpers or redirected care, while those with*

territory shortage may involve either retained offspring or plural breeding systems where multiple females build nests in the same territory (see Supplementary Data 1 for details). The limited number of species with each of these social systems is not adequate to test for statistical differences in kin discrimination and helping effort but, compared to species with territory shortage, the degree of kin discrimination was relatively high and helping effort was relatively low for species in which there is no territory shortage, regardless of the social system (Kin discrimination (mean correlation coefficient \pm SE): species with retained offspring: 0.14 ± 0.10 , $n = 11$ species; plural breeding: 0.07 ± 0.06 , $n = 2$ species; colonial: 0.49 ± 0.10 , $n = 4$ species; redirected care: 0.57 ± 0.20 , $n = 3$ species; see Supplementary Data 1). Helping effort (average \pm SE compared to breeders): species with retained offspring: $89 \pm 10\%$, $n = 24$ species; plural breeding: $88 \pm 13\%$, $n = 4$ species; colonial: $63 \pm 7\%$, $n = 5$ species; redirected care: $56 \pm 13\%$, $n = 6$ species; immature helpers: $53 \pm 1\%$, $n = 2$ species; see Supplementary Data 1). This indicates that the reported overall effects were not driven by species with a particular social system either for kin discrimination. '.

Comment 18. L63. There are alternative models—things like reproductive skew theory—which are based on the idea of kin selection but incorporate habitat saturation. They make predictions about helping effort (e.g. Figure 1B) in relation to habitat constraints. Your results for this analysis seem to be consistent with some of these models. For example, helping effort will be higher in species where independent breeding is more constrained, and not necessarily because of the potential for territory inheritance. In other words, can you explain this result in the absence of territorial inheritance being the driving force with something that may be more parsimonious? I realize that Figure 2B at least partially ameliorates this concern, but are the prospects of territorial inheritance higher because the constraints on independent breeding are so much higher? In other words, are the observed relationships ultimately related to habitat saturation, delayed dispersal, etc. and only secondarily to territorial inheritance.

Response: The idea to discuss the results in a reproductive skew theory setting is an excellent suggestion. By focussing mainly on benefits of helping (rather than the benefits of staying) I had somewhat overlooked this. Indeed, thinking about it this way makes clear that helping effort may to some extent be higher because of other benefits (like high survival in a resident territory) associated with habitat saturation and the lack of outside options (reflecting subordinates being 'forced' to pay more to stay in more saturated habitats). I have now discussed the results in this light, also making important references to skew and biological market theories. I think that this combination of predictions from these theories as well as 'pay-to-stay' and 'group augmentation' hypotheses indeed broadens the scope of the paper.

Line 141-170: *'Why do subordinates invest in feeding unrelated individuals if they live in a territory where they may reproduce in the future? Several complementary mechanisms are probably involved, based on benefits of group living that operate immediately or in the future, alongside or in place of kin selection'²¹. Individuals may be more likely to stay and willing to 'pay' more in helping if the benefits of staying (e.g. territory inheritance) are larger ("pay-to-stay"²⁰). The result that increasing prospects of territory inheritance lead, irrespective of kin-selected benefits, to higher helping effort seems to suggest that this is indeed the case. However, if inheritance rates reflect the degree of habitat saturation and the lack of options for subordinates to survive independently outside their resident territory, it could also be that breeders in species in highly saturated habitat can afford to force subordinates to help more, since subordinates are not able to leave successfully (as predicted by biological market theory¹⁹ and related to reproductive skew models²²). If individuals indeed pay more to stay if outside options are limited, higher territory*

*inheritance rates per se would not necessarily be the cause of higher helping effort but both would rather be the effect of delayed dispersal and the lack of outside options. It remains to be tested whether subordinates are forced to pay more when constraints for independent breeding are more severe. However, in addition to, or regardless of pay-to-stay motivations for helping, if helping leads to larger groups (as is the case in many cooperative breeders²³) a high prospect of territory inheritance itself will also promote helping behaviour for a number of potential reasons²¹. Larger groups are more stable and/or better able to defend the territory in many species²⁴; this facilitates territory persistence and improves the chance of individuals inheriting the territory. Moreover, larger groups may expand the territory so that subordinates can split off a part of it²⁵; a common route to independent breeding in some species (e.g. laughing kookaburras *Dacelo novaeguineae*²⁶, Florida scrub-jays *Aphelocoma coerulescens*²⁷). Additionally, helping as a subordinate may lead to improved future breeding success after becoming a breeder in the territory because the resulting augmented group contains future helpers²⁸. Thus, regardless of the mechanism, if individuals can inherit the territory and queues for inheritance are stable (as is usually the case²¹), mutualistic and reciprocal benefits provided by newly-recruited group members can maintain cooperation in a self-reinforcing way.'*

Moreover, I have made other sections a bit broader, in order to not rule out 'pay-to-stay' as explanation for the results:

- L. 39-50: *'In many cooperative breeders, shortage of suitable territories ('habitat saturation') limits subordinates' opportunities for independent reproduction^{1,16} and theory predicts that both the lack of outside options and the prospects of territory inheritance may explain why such subordinates stay in a group and help^{15,17-19}. Importantly, helping behaviour may facilitate survival and ultimate territory inheritance because helpers avoid aggression and eviction by breeders ('pay-to-stay' hypothesis^{19,20}) or contribute towards the establishment of larger cooperative groups that improve survival, territory defence, group stability or the ability to expand and split the territory ('group augmentation' hypothesis^{5,15,21}). Despite this clear theoretical prediction and the fact that territory inheritance is a common and important route to independent breeding in many species²¹, it remains unclear whether habitat saturation and prospects of territory inheritance can explain helping behaviour, especially by unrelated individuals who do not gain kin-selected benefits.'*

- L. 182-184: *'The finding that direct benefits of philopatry and territory inheritance can predict helping behaviour where kin selection cannot has substantial implications for our understanding of helping behaviour, group living, and cooperation in general.'*

I have added the following references to the reference list:

Grinsted, L. & Field, J. Market forces influence helping behaviour in cooperatively breeding paper wasps. *Nat. Comm.* 8, 13750 (2017)

Johnstone, R. A. & Cant, M. A. Models of reproductive skew: outside options and the resolution of reproductive conflict. In *Reproductive Skew in Vertebrates: Proximate and Ultimate Causes* (eds Hager R. & Jones C. B. 3-23 (Cambridge University Press, 2009)

Lehmann, L., Perrin, N. & Rousset, F. Population demography and the evolution of helping behaviors. *Evolution* 60, 1137-1151 (2006)

Comment 19. L223. I have a few questions about your choice of species to include or exclude. In your definition, you are effectively limiting your dataset to primarily singular breeders and excluding

many plural breeding species (brown jay, chestnut-crowned babbler) and joint-nesting species. While I don't take issues with this, many of the species you did decide to include in the dataset are plural breeders and perhaps seem to meet these criteria (bell minor, Galapagos mockingbird, sociable weaver, white-winged chough, etc), especially if you varied that >10% threshold. I think doing some analyses to look at these factors (e.g., plural versus singular breeding) with the species you included in your dataset will be useful and informative. If subordinates have options to breed within the group, then that might also explain helping non-relatives beyond territorial inheritance. In fact, this seems like a very plausible alternative hypothesis that you could test with your data.

Response: I agree that in several species subordinates may help because they can obtain parentage. I have excluded all species in which co-breeding/joint reproduction takes place, since often it cannot be determined whether 'helpers' reproduced or not in the reproductive attempt for which helping was assessed (and helping behaviour does not reflect altruism in such species, therefore falling outside the scope of this study). I want to highlight here that I only used helpers (individuals that do not reproduce in the territory but may breed in the territory in the future) to assess helping behaviour in plural breeding species. I mention this now in L. 215-219: *'In many cooperative breeders, subordinates are retained individuals who delayed dispersal and reproduction, and remained in a territory as a consequence of habitat saturation (a shortage of vacant breeding territories due to a lack of suitable breeding habitat; including plural breeding species in which multiple females may reproduce in independent nests in a territory).'*

As mentioned in comment 17 above (splitting up species categories), I had initially considered separately addressing plural breeders for the analyses in figure 1 (multiple females having an own nest within a defended territory), but decided against further splitting up of the analyses due to the limited availability of data for plural breeding species. Like for comment 17, I have however reported the degree of kin discrimination and helping effort in L. 94-111 (see comment 17 for quote and sample sizes).

Comment 20. L269. In many species, non-breeding subordinates perform other important roles like nest defense. What happens to your analysis of helping behavior if you remove these individuals and only consider individuals that actually bring food to the nest?

Response: I agree and have, also following suggestions by reviewer 1 (comment 10), conducted an analysis excluding the non-helping subordinates from the calculation of 'helping effort'. I have now provided these analyses as Supplementary Tables 2b and 4b, and added to the main text that excluding non-helping subordinates did not change the results (see comment 10 for quotes).

Comment 21. Appendix. It would help to make sure the variables in the models/tables have the same names in the appendix. For example, "habitat saturation" should be the same as "territorial shortage", as should "rate of extra-pair paternity" and "% nests with extra-pair young", etc. By using different names in your appendix, it becomes harder to figure out which data are being used in which models.

Response: This is a good idea. I have now made the terms consistent across the manuscript, tables and the Supplemental Data.

REVIEWERS' COMMENTS:

Reviewer #1 (Remarks to the Author):

I find this manuscript to be much improved in terms of clarity of what was measured and why. I have the following comments on the revised manuscript:

L 66: delayed dispersal does not always imply territory shortage. Delayed dispersal may occur for a number of reasons, and will be present in some species even when there are vacant territories to fill.

L 93: rephrase to clarify: 'extra-pair paternity rates did not change the results'

L 106-111: what do the numbers here represent? Given as percentages, but % of what? The numbers are so vastly different between the breeding systems that I wonder if there was some error here and they are reported on different scales?

L 112: grammar error here? Sentence structure odd

L 119: huge range! This large range made me refer to the range for the other breeding system on L 114 for comparison, but no range given? Please add in

L 114 & 119: as for comment further up, these values are given on a different scale, making a direct comparison between them more difficult. For the first, 10% is initially represented as 0.10, whereas for the latter system, it is represented as 31%.

L 159: there is actually very little empirical evidence of pay-to-stay strategies occurring in cooperatively breeding birds: this may add weight to your argument here.

L 190: reference needed for statement regarding kinship explaining helping patterns

L 191: reference needed for this statement about direct benefits being more imp't in many other species

L 222: is there really any convincing evidence that plural breeding is a result of territory shortage? Does plural breeding not still occur even when there are territory vacancies?

Reviewer #2 (Remarks to the Author):

The author has done a nice job of revising the manuscript. The analyses are now more thorough and the presentation of the data are more transparent. The trends reported previously are now more believable and robust. I am also happy to see consideration of a more nuanced interpretation of the results, and the incorporation of alternative hypotheses. Overall, I am satisfied with the revision and have no major comments. My only minor issue

is that the paragraph beginning P7L141 is quite long and cumbersome. I would like to see that broken up and revised a bit. Otherwise, I'm happy with the MS as is.

Second Revision NCOMMS-17-10999-A “Direct benefits explain interspecific variation in helping behaviour among cooperatively breeding birds”

Response to reviewers

I thank the reviewers for their insightful comments throughout the review process. I am very happy to hear that both reviewers agree that the writing and analyses have improved a lot in terms of clarity. I have incorporated the changes based on their last comments and address below each comment how and where I made changes to the manuscript (line numbers refer to line numbers in the revised version of the manuscript including ‘All markup’ track changes).

Reviewer #1 (Remarks to the Author):

I find this manuscript to be much improved in terms of clarity of what was measured and why. I have the following comments on the revised manuscript:

L 66: delayed dispersal does not always imply territory shortage. Delayed dispersal may occur for a number of reasons, and will be present in some species even when there are vacant territories to fill.

Response: This is correct. I have changed this sentence so that it also better describes the reasoning for using territory shortage and so that the definition corresponds to the definition in the methods section. L. 76-79 now reads: ‘I subsequently compared these measures between species with territory shortage (i.e. species in which independent breeding by subordinates is constrained by a shortage of vacant territories for independent breeding) and those without (i.e. colonial species, species with redirected care and species with immature helpers).’.

L 93: rephrase to clarify: ‘extra-pair paternity rates did not change the results’

I have rewritten this sentence (l. 109-110: ‘However, including extra-pair paternity rates did not change the results (Supplementary Table 2c).’)

L 106-111: what do the numbers here represent? Given as percentages, but % of what? The numbers are so vastly different between the breeding systems that I wonder if there was some error here and they are reported on different scales?

Response: I realised that it was indeed unclear what the percentages refer to. I have now rewritten this part to clarify and make the definition of ‘helping effort percentage’ the same as that in the figure legends: l. 124-125: ‘Helping effort (mean \pm SE percentage offspring food provisioning per helper, relative to breeders):’. Please note that the ranges are correct (for comparison see Figure 1b).

L 112: grammar error here? Sentence structure odd

Response: I have added ‘or for helping effort’ to l. 130 which was missing from this sentence

L 119: huge range! This large range made me refer to the range for the other breeding system on L 114 for comparison, but no range given? Please add in

Response: I have now also provided the range of r^2 values for species with territory shortage. L. 131-133: ‘The mean coefficient of determination (r^2) between helping and relatedness in species with territory shortage was 0.10 ± 0.03 (range = $0.0004 - 0.36$)...’.

L 114 & 119: as for comment further up, these values are given on a different scale, making a direct comparison between them more difficult. For the first, 10% is initially represented as 0.10, whereas for the latter system, it is represented as 31%.

Response: I have now made this more comparable by stating the mean r^2 in l. 139-140:

'relatedness explains on average 31 ± 10% (range = 6-93%) of the variation in helping behaviour (i.e. the mean coefficient of determination (r^2) between helping and relatedness was 0.31).'

L 159: there is actually very little empirical evidence of pay-to-stay strategies occurring in cooperatively breeding birds: this may add weight to your argument here.

Response: This is a good point. I have added this argument to l. 181-191: *'Since only few studies have tested the pay-to-stay hypothesis in cooperatively breeding birds with mixed results^{14,20,23,24}, it would be worthwhile to test whether subordinates are indeed forced to pay more when constraints for independent breeding are more severe.'*

I have added the reference to Mulder and Langmore 1993.

L 190: reference needed for statement regarding kinship explaining helping patterns

Response: I have added a reference to Griffin and West (2003) who nicely display which species have kin-biased helping.

L 191: reference needed for this statement about direct benefits being more imp't in many other species

Response: I added references here (l. 230) for species in which direct benefits of help have been suggested: *'While helpers clearly discriminate based on kinship in some species (see⁷), direct fitness benefits appear equally, if not more, important in explaining helping behaviour in many others (e.g.^{14, 29, 36}).*

L 222: is there really any convincing evidence that plural breeding is a result of territory shortage? Does plural breeding not still occur even when there are territory vacancies?

Response: In all the plural breeding species included here (Australian magpie, Galapagos mockingbird, red-winged fairy-wren, splendid fairy-wren), individuals indeed live in permanent territorial groups, habitat is saturated (territory shortage) and helpers delayed dispersal and remain in their resident territory. Plural breeding may, in itself, not result from territory shortage, but helping behaviour in such species most likely does.

see for references:

***Red-winged fairy-wren:* Rowley et al. 1988. Emu 88, 161-176;**

***Galapagos mockingbird:* Kinnaird and Grant 1982, Behav Ecol Sociobiol 10, 65-73.**

***Splendid fairy-wren:* Rowley 1981, Z. Tierpsychol. 55, 228-267.**

***Australian magpie:* Durrant, K. L. The Genetic and Social Mating System of a White-backed Population of the Australian Magpie (*Gymnorhina tibicen tyrannica*) (thesis, Griffith University, 2004)**

Reviewer #2 (Remarks to the Author):

The author has done a nice job of revising the manuscript. The analyses are now more thorough and the presentation of the data are more transparent. The trends reported previously are now more believable and robust. I am also happy to see consideration of a more nuanced interpretation of the results, and the incorporation of alternative hypotheses. Overall, I am satisfied with the revision and have no major comments. My only minor issue is that the paragraph beginning P7L141 is quite long and cumbersome. I would like to see that broken up and revised a bit. Otherwise, I'm happy with the MS as is.

Response: Thank you very much for the positive assessment. I agree that part of the discussion was dense in places and I have split up this paragraph and revised it to make it better readable. The changes are highlighted in l. 166-222 with track changes.